# Computer Vision Estimation of Physical Parameters and Its Application to Power Requirements of Natural and Artificial Swimmers

**Michael Coe \*,† and Stefanie Gutschmidt †**

Department of Mechanical Engineering, University of Canterbury, Christchurch 8140, New Zealand; stefanie.gutschmidt@canterbury.ac.nz

\* Correspondence: michael.coe@pg.canterbury.ac.nz

† These authors contributed equally to this work.

**Abstract:** A useful measure of efficiency of transport in aquatic animals and autonomous underwater vehicles is cost of transport. Often, cost of transport data on specific animals or platforms is not readily available or does not fit specific use cases, but images are readily available. In this work, we present a methodology to synthesize such data without the need for a specimen or laboratory tests. We propose a computer vision in a methodology called Ika-Fit to determine important physical characteristics, such as surface area, slenderness ratio, and mass, that are used for a cost of transport model. The Ika-Fit method provides a good estimation of parameters when compared to biological data and robotic platforms. These parameters are estimated for existing engineered systems, and the model is compared to published data; the model is found to demonstrate higher accuracy using fewer parameters in estimating cost of transport over existing methods.

**Keywords:** cost of transport; power estimations; autonomous underwater vehicles; computer vision applications; surface area

## 1. Introduction

Can we make a robotic fish or autonomous underwater vehicle (AUV) that competes with the performance of nature? When building an AUV, what is the best locomotion mode to use for a specific mission application? Cost of transport (CoT) offers a way to more holistically and accurately compare multiple systems that operate with different locomotion modes, sizes, and masses. The CoT metric is a measure of the energy efficiency of a biological animal or engineered system in transporting a unit mass over a unit distance [1]. In biological systems, this metric is a combination of the base metabolic rate and the active metabolic rate [2]. For engineered systems, AUVs for example, this metric indicates the amount of energy that is expended for a specific mission time [3,4]. Similarly, CoT can be used to compare energy efficiency of multiple locomotion modes [5], fish species [6], or AUVs [3,4,7].

Many groups have performed water tunnel tests on biological swimmers [5,8,9] in order to determine CoT. The Patankar group at Northwestern University compiled this data and included data on flying animals to develop an allometric scaling relationship between the animals' mass and CoT [6]. The group found that CoT has an inverse relationship with respect to unit mass in swimming animals. Performing a similar analysis on engineered systems is difficult as there is very little published CoT data. Many AUVs are proprietary, and there is a lack of published research articles that include CoT calculations or thrust tests. The same is true for bio-mimetic platforms. Out of 139 platforms surveyed, only 3 groups reported CoT results.

The first and only group to attempt an allometric analysis of CoT in AUVs is the combined Murphy group of Newcastle University and Phillips group at the University

of Southampton as part of the Nature in Engineering for Monitoring Oceans (NEMO) project [3,4]. This analysis used a simplified CoT model on a small set of conventional AUVs and sea gliders, but did not include bio-inspired robotic platforms. Interestingly, their analysis found that conventional AUVs have a lower CoT when compared to biological animals for the same displacement. This analysis was later used by Fish at Westchester University to argue that conventional AUVs may be more energy efficient, but biological animals and bio-mimetic robots offer increased maneuverability and stealth [7]. This raises the question of when it is appropriate to choose a bio-mimetic platform over a conventional AUV. This research seeks to extend the analysis by Phillips [3] to bio-mimetic robotic platforms to obtain a clearer picture of underwater vehicle designs and their performance for a specific mission task.

One approach is to develop a scaling relationship for propulsion power depending on the locomotion mode. The Mahadevan group at Harvard University proposed a new dimensionless factor that is a combination of the Reynolds and Strouhal number called the swim number $Sw$ [10]. They analyzed over 3000 larvae, amphibians, fish, reptiles, birds, and mammals and found that the Reynolds number scales with $Sw^{\frac{4}{3}}$ in the laminar regime and linearly in the turbulent regime. The group did not include thrust or CoT in their analysis. Yu and Huang developed a scaling relationship between thrust and undulatory parameters such as wave speed, wavelength, and amplitude [11]. Their analysis showed that the mean thrust depends on $St^2$ and the relative motion $(1 - U/c)$, where $U$ is the reference velocity and $c$ is the undulatory phase speed. This study relied on the perfect fluid-structure interaction of a fish analogue and did not include the losses from different actuation methods, such as linkages and motors versus muscles. Furthermore, both these studies are valid for animal locomotion, while we require a method that also includes conventional propeller-driven AUVs and bio-mimetic robots in the analysis. There is currently no study that combines conventional AUVs, biological fish, and engineered bio-mimetic robots in the framework of the cost of transport metric. This research offers a methodology (Ika-Fit method) that can successfully compare platforms with different locomotion modes when no CoT data is available to determine an appropriate platform for a given application.

The present paper is organized as follows: Section 2 (The Cost of the Transport Metric) describes the formulation of the CoT model. Section 3.1 (Ika-Fit Method) describes the surface area and volume approximation method. Section 4 (Validation) compares the methodology with a 3D model that has been scaled and with published biological data. This section also compares the Ika-Fit Method to the methods described in [3,12–14] and validates the methodology against bio-mimetic robotic platforms where surface area was published. Section 4.3 (The Application to CoT) section shows the method being applied to biological animals, bio-mimetic robots, and AUVs using (5) and, furthermore, shows how it compares to published CoT data. This section also discusses where improvements to the algorithm and CoT model can be made and provides the pros and cons of the algorithm and CoT models.

## 2. The Cost of Transport Metric

The commonly accepted equation for CoT was first derived by Schmidt–Nielsen [2] and is given as follows:

$$CoT = \frac{AMR}{U} \tag{1}$$

$$CoT = \frac{AMR}{M \cdot U} \tag{2}$$

where AMR is the active metabolic rate, $M$ is mass, and $U$ is the speed of the animal. Equation (1) is commonly used in biological texts, as (2) is the mass-normalized form and is more common for engineering applications.

For animals, obtaining the direct AMR is evasive and difficult to measure; therefore, oxygen uptake $\dot{M}_{O_2}$ is used and measured by a respirometer in either a circular tank, pond [15] or a water tunnel [9,16]. $\dot{M}_{O_2}$ for animals is generally given in units of $(mgO_2kg^{-1}min^{-1})$, which can be converted to metabolic power $(P_M[W])$ by assuming that all the oxygen is converted to energy, with the conversion factor from $mgO_2$ to J given by Elliot and Davison [17]:

$$\text{conversion factor} = 14.14 \ \frac{J}{mgO_2}, \tag{3}$$

which gives a metabolic power of:

$$P_M = 0.2357 \ \dot{M}_{O_2} \cdot M, \tag{4}$$

where 0.2357 is the conversion factor, (3), divided by 60 to convert the minutes in $\dot{M}_{O_2}$ to seconds.

Fluid dynamicists have approached calculating CoT separately using drag theory by calculating the amount of thrust that an animal would need to overcome the viscous drag of the fluid [2,18,19]. This approach has the benefit of not needing physical laboratory testing of the animal to measure $\dot{M}_{O_2}$ at different swimming velocities. Using this model, the CoT for the animal can be expressed as a base metabolic power $(P_B)$, a propulsive power $(P_P)$, a mass, and velocity [3,4,12]:

$$CoT = \frac{P_B + \frac{1}{2\eta_a\eta_p}\rho C_D A_s U^3}{M \cdot U}, \tag{5}$$

where $P_B$ is found by extrapolating $\dot{M}_{O_2}$ to $U = 0$ to obtain the power at 0 velocity and converted using the conversion factor, (3), and $\eta_a$ and $\eta_p$ are the unit-less actuator and propulsive efficiency, respectively. $\eta_a$ represents the efficiency of the linkages and actuation mechanisms, such as motors, shafts, and couplings. $\eta_p$ represents the efficiency of the propulsion mode, such as the propeller efficiency for conventional AUVs or the flapping propulsor efficiency for bio-mimetic robots. $\rho$ in (5) is the fluid density in $kgm^{-3}$, $C_D$ is the unit-less drag coefficient, $A_s$ is the wetted surface area in $m^2$, and $U$ is the free-stream velocity in $ms^{-1}$.

In practice, $C_D$ is a measure of towed resistance calculated using the resistance procedure outlined by the International Towing Tank Committee (ITTC) [20]. This recommended procedure fits the drag coefficient to an empirical line using the equation:

$$C_D = (1 + k)C_{FM} = (1 + k)\frac{0.075}{\left(\log_{10} Re - 2\right)^2}, \tag{6}$$

where $(1 + k)$ is the form factor defined by Hoerner [21] and $Re$ is the Reynolds number. The form factor is based on the slenderness ratio

$$SR = \frac{L}{D}, \tag{7}$$

where $L$ is the length of the object and $D$ is the diameter of the cross section. The full form factor is given as [20,21]:

$$FF = (1 + k) = 1 + 1.5 \ SR^{3/2} + 7 \ SR^3. \tag{8}$$

From inspection, CoT changes significantly with the subject's physical dimensions. Specifically, the wetted surface area is directly proportional to propulsive power. From Hoerner [21], the drag coefficient, (6), is directly proportional to the physical parameters in

the slenderness ratio in (7). If it is assumed that the animal is neutrally buoyant, then the physical dimensions effect mass as well.

To further elucidate the importance of using accurate physical parameters when using the CoT model, (5), several parameters are varied with all other parameters held constant and are shown in Figure 1. Regions in Figure 1 are divided by a vertical dashed line with region 1 on the left hand side and region 2 on the right hand side. The demarcation of these regions are marked by the convergence and divergence of the lines on either side of the demarcation point.

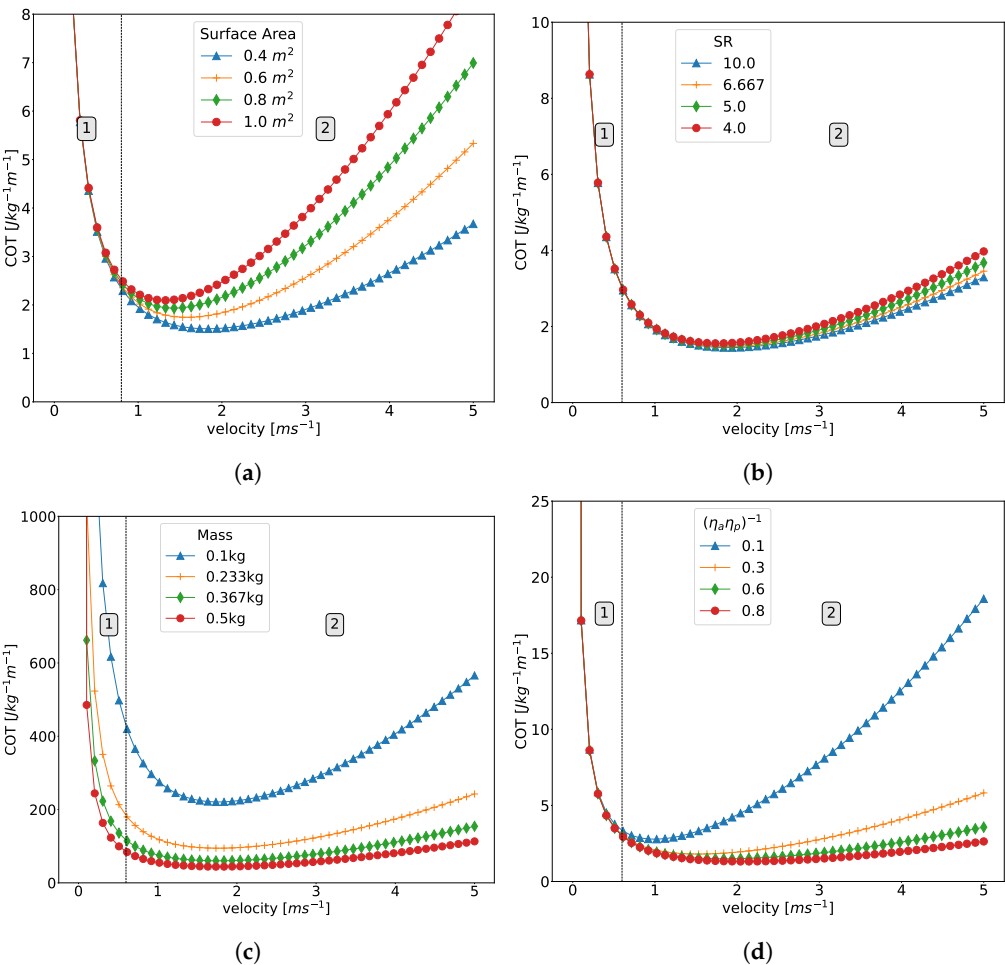

**Figure 1.** (**a**) The influence on CoT of changing the (**a**) wetted surface area, (**b**) slenderness ratio, (**c**) mass, and (**d**) $1/(\eta_a \eta_p)$ with all other variables being equal. The dashed black line represented the division between the region in which hotel power dominates at low velocities and the region where propulsion power dominates at higher velocities. The division line is placed at $0.6~\text{ms}^{-1}$ for all graphs.

Of important note is that $P_B$ dominates at lower velocities (region 1), while $P_P$ is dominant at higher speeds (region 2), which gives the characteristic U-shaped curve [3]. Varying the wetted surface area will shift the right side of the graph upwards, causing a higher CoT in region 2. Shifting (7) effects (8), which causes an increase in (6). This causes the CoT to increase in region 2, but the effect is much less than that of the surface area. Decreasing mass in (5) will shift the denominator, which will cause the CoT to increase throughout the entire CoT curve. Finally, decreasing efficiency will increase the CoT in region 2, since it is inversely proportional to propulsion power.

There is very little information giving allometric relationships for length/mass versus surface area of various aquatic animals, AUVs, or artificial swimmers. The animal-based literature mainly focuses on aquaculture and the colonization of lice on farmed fish.

The current method for accurate measurement of the surface area of fish is an adapted wrap method [22]. The animal is anesthetized, the fins are dissected, and the body of the fish is wrapped in paper. The paper is cut such that all the edges are flush with each other. Once cut, the paper is laid flat on graphical paper where the surface area is measured [22–24]. Another method is to wrap the body with cotton strings at 2 mm increments along the body length [25]. The strings are then measured and translated to 2D coordinates to get the surface area. Similarly to the method described by O'Shea et al., this method has some interpolation error associated with it and is not as accurate as the previous wrap method. In the realm of engineering systems, there have not been any studies that show allometric relationships between length/mass and surface area.

There has been considerable research done in the realm of fish classification and estimation of physical dimensions using computer vision. Traditional image segmentation and volume estimation methods are presented by Siswantoro et al.; these authors used k-means clustering and the Sobel operator [26]. Balaban et al. measured Alaska pollock (Theragra chalcogramma) by taking side and top view images and then estimating the body contour as a b-spline. This was then used to calculate the volume [27]. In contrast, Rantung et al. measured the length, height, and width through side and top views of the fish and used these measurements as inputs that divide the fish into discrete elliptical discs [14]. Prior to using this algorithm, the camera was calibrated such that the length and height of each pixel was known.

With the recent increase in computational power, there has been increased emphasis on the use of convolution neural networks (CNN) for this task. Yang et al. provides a review of the use of deep learning techniques in fish farming [28]. The group shows that using CNNs for image segmentation and estimation of fish parameters can achieve between 0.2% and 5% accuracy. A disadvantage of these models is that they require a lot of data to train and are less accurate when trained with a limited data set. Additionally, these models are valid only for the species they are trained on [28–30]. While there are large datasets available to train CNNs [31], this study's focus is on bio-mimetic robots and AUVs, which are not always shaped as fish. For this reason, we elected to employ a simpler model that gives complete manual control over determining the contours and fit parameters versus being a "black box". Furthermore, these methods are not strictly used to measure fish surface area as would be needed for the CoT model.

Few attempts have been made to measure fish surface area using computer vision techniques which involve the scanning of fish into a computer and then measuring their dimensions. Balaban et al. measured Alaska pollock (Theragra chalcogramma) by taking side and top view images and then estimating the body contour as a b-spline. This was then used to calculate the volume [27]. In contrast, Rantung et al. measured the length, height, and width through side and top views of the fish and used these measurements as inputs that divide the fish into discrete elliptical discs [14]. Prior to using this algorithm, the camera was calibrated such that the length and height of each pixel was known.

A method not based on computer vision is derived by Murphy and Haroutunian and requires the length and mass of the animal or engineered system. This method derives an equivalent diameter as an input to a prolate spheroid approximation used in calculating the surface area [3,12,13]. This particular approach is beneficial because much of the literature on fish species only provide their length and mass. Rantung et al. also presents this prolate spheroid method, but without the equivalent diameter derivations [14].

The contact methods described by O'Shea et al. and Ling et al. require the animal to be physically present and anesthetized. In many cases, obtaining a specimen is difficult, and applying this method to AUVs and artificial swimmers is impractical. The contact-less methods described in Murphy and Haroutunian and Rantung et al. are more appropriate when trying to synthesize data from specimens that are not physically present. These methods have the drawback that some of the physical dimensions, such as length, mass, width, and height, are needed beforehand to obtain a relatively accurate measurement.

Concerning efficiency in biological animals, actuator efficiency is a combination of adenosine triphosphate (ATP) conversion and muscle efficiency [32]. Conventional propeller underwater vehicles and bio-mimetic robots have an actuator efficiency that is a combination of the actuator itself and the linkages linking the actuator to the propulsor. Propulsion efficiency is dependent on the type of propulsion employed; this encompasses body or fin undulation for biological and bio-mimetic models, and either buoyancy or propeller propulsion for conventional AUVs [33–36]. A summary of the typical efficiencies ($\eta_a$, $\eta_p$) are given in Phillips et al. [3] and is expanded upon in Table 1.

**Table 1.** Table of typical efficiencies summarized in [3] and expanded on with data from a review in [37].

| Actuator Type | Typical Efficiencies | Reference |
|---|---|---|
| Direct current motor | 0.60–0.90 | [38] |
| Pneumatic cylinders | <0.67 | [39] |
| Dogfish red muscle | <0.51 | [32] |
| Dogfish white muscle | <0.41 | [32] |
| Diesel engine | <0.40 | [40] |
| Bluegill sunfish | 0.26–0.37 | [41] |
| Electroactive polymers | <0.38 | [37] |
| Shape memory alloys | <0.10 | [37] |
| Nanoparticle-based | <0.01 | [37] |
| Twisted coil polymer | 0.10–10 | [37] |
| Ionic polymer metal composites | <3.0 | [37] |
| Dielectric elastomer | <90 | [37] |
| Conducting polymer | <18 | [37] |
| Piezoelectric | 90 | [37] |
| **Propulsor Type** | **Typical Efficiencies** | **Reference** |
| Buoyancy engine | <0.50 | [34] |
| Propeller | <0.53 | [42] |
| Biological | 0.80–0.90 | [43] |

## 3. Materials and Methods

The method outlined in this research is called the Ika-Fit method. "Ika" is the Māori word for fish, and the research involves using computer vision to fit contours to various natural and artificial fish.

### 3.1. Ika-Fit Method

This research differs from previous techniques, described in Section 1, in that the curvature of the specimen's body is fit to a 6 degree polynomial, a National Advisory Committee for Aeronautics (NACA) airfoil, and the top and bottom of the animal is treated as separate ellipse partition discs. This formulation allows for a more accurate estimation of the physical parameters, discussed in detail in Section 4. This is in contrast to existing methods that treat the fish-like body as a prolate spheroid where the surface area can be readily calculated or as an ellipse disc based on the height and width of each partition.

Side and top images of fish were obtained from the digital fish library project [44]. Image artifacts were manually removed from the black background and two copies of the side view, one with fins and one with fins removed, provided only body geometry, as shown in Figure 2. For validation purposes, a king salmon (Oncorhynchus tshawytscha) was laser scanned into a mesh using a Kreon Ace-7-30 laser measurement arm (KREON Technologies, Limoges Cedex, France). This scanned data was imported into the open-source software Blender where it was scaled in the interval [0.1:100], and the surface area and volume were readily evaluated. Figure 2 shows the images used for the scanned salmon, but all images used in this research follow the same format.

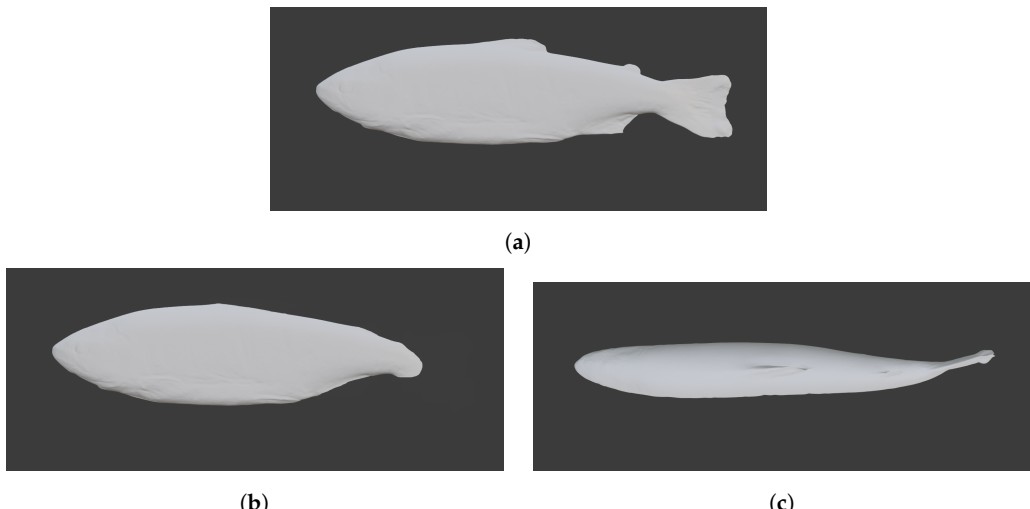

(**a**)

(**b**)　　　　　　　　　　　　　　　　　(**c**)

**Figure 2.** Images used in the validation of the Ika-Fit method with salmon modeled in Blender. (**a**) Salmon with caudal fin, (**b**) Salmon with no caudal fin, and (**c**) Salmon top with fins.

### 3.1.1. Image Segmentation and Contours

The Python OpenCV package was used to process the images and determine body geometry contours [45]. The image segmentation algorithm used was an adapted version of the automatic image segmentation algorithm given in Siswantoro et al. [26], and an overview of the adapted algorithm is shown in Figure 3.

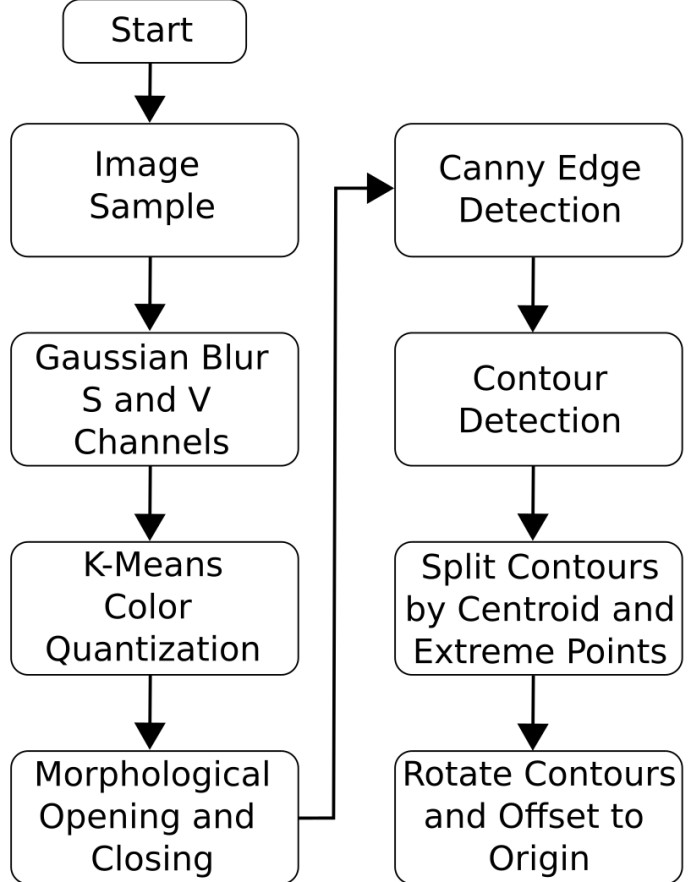

**Figure 3.** The steps taken for image segmentation.

When an image is read into the program, it is read as a matrix with dimensions $(H, W, C)$, where $H$ and $W$ are the height and width of the image and $C$ represents a vector of three color channels (blue, green, red) as in Figure 4a. The image is converted from RGB (red, green, blue) to HSV (hue, saturation, value), where a Gaussian blur is used on the S and V components before merging the channels back to an image shown in Figure 4b,c. K-means color quantization is performed to threshold the image into two colors, as in Figure 4d. Morphological opening and closing on the image is used to fill any gaps in the fish shown in Figure 4e. Canny edge detection is used to determine the edges where the fish foreground meets the background as in Figure 4f.

Shape extraction is used on the canny edge detection image using the find contours method in [45], which implements the algorithm presented in [46]. The algorithm works on a binary image with the background being black pixels (255) and the foreground being white pixels (0). The first foreground border pixel is found and the surrounding pixels are checked for being black or white. In the case of multiple white pixels, the outside pixel is chosen and the window is moved to that pixel. The contours represent $(x, y)$ coordinates of continuously connected white pixels. The final segmented image is shown in Figure 4g, where the contour is shown over the image after morphological operations had been performed.

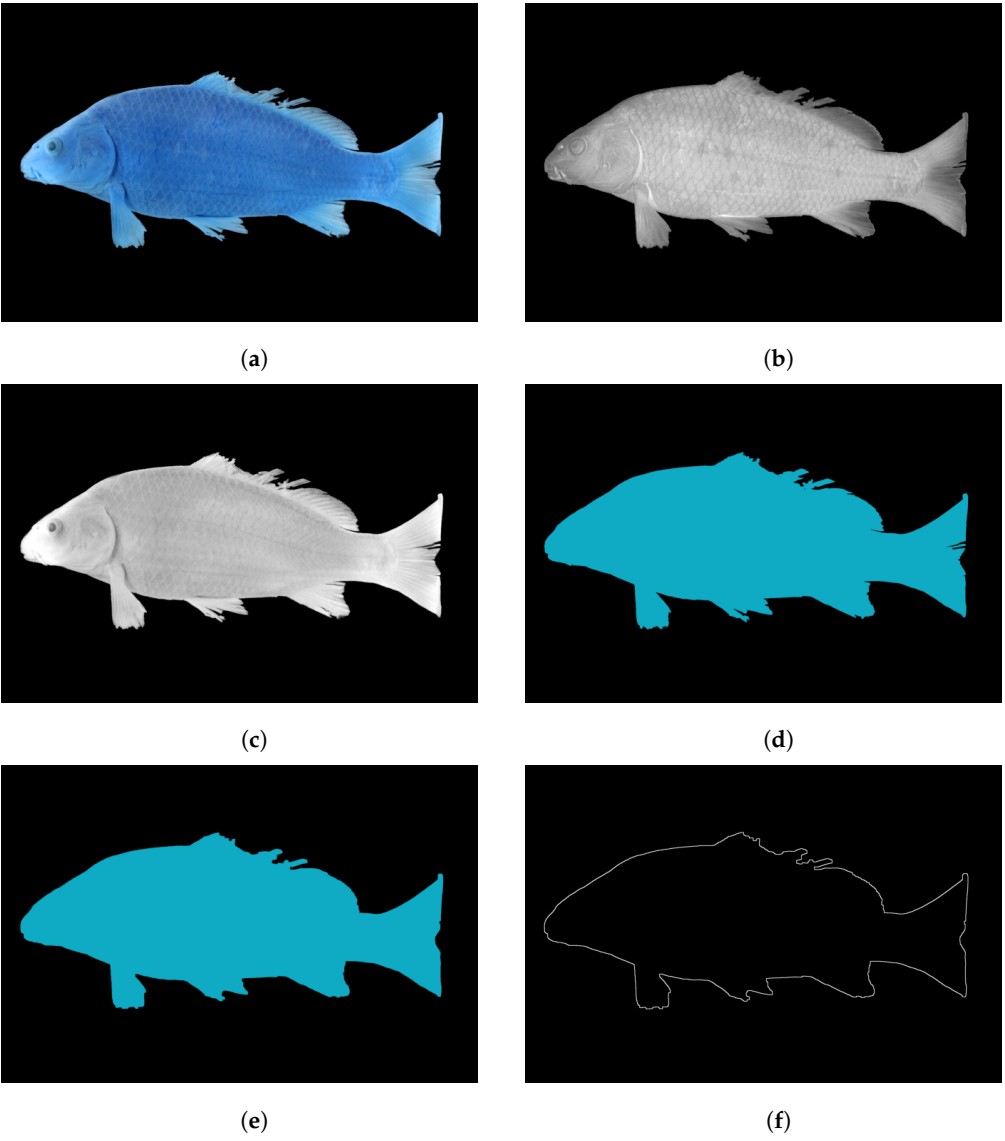

**Figure 4.** *Cont.*

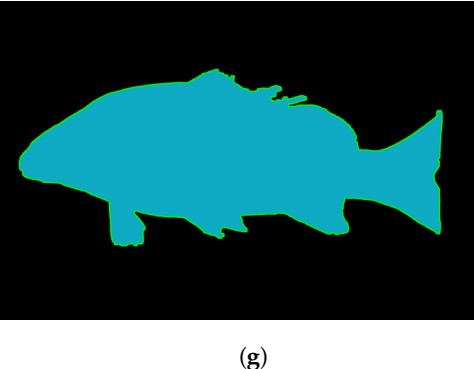

(**g**)

**Figure 4.** Images showing the algorithm on a complex fish specimen. (**a**) Initial image, (**b**) S and (**c**) V components, (**d**) K-means color quantized image, (**e**) image after morphological opening and closing, (**f**) canny edge-detected Image, and (**g**) contours overlayed over image (**e**).

The centroid and extreme points of the contour were used as a dividing line to divide the specimen into a top half and bottom half, as shown in Figure 5. The Figure shows the resulting contour in white around the original image. The blue line represents a line that goes through the centroid of the specimen to the x-coordinates of the extreme left and right points called the mid plane. The red line represents the line used to divide the specimen by the left and right most extreme points. This shows the difference between partitioning the specimen using the midline versus the dividing line used in this research. The contours above and below the divide line are separated into the top and bottom hull, respectively. This separation of the contour into two hulls allows the algorithm to treat the top and bottom of the specimen as different ellipsoid discs.

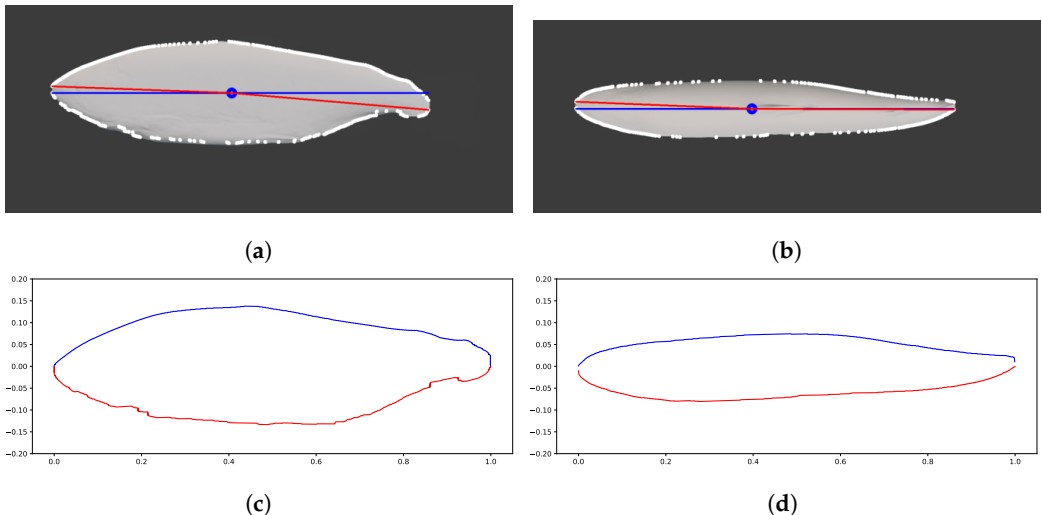

**Figure 5.** (**a**,**c**) Side and top view with the overlayed centroid (blue dot), line slicing through the middle of the specimen (blue line), dividing line (red line) to extrema points, and white line representing the contours. (**b**,**d**) Plotted contour after dividing the top and bottom of the specimen and zeroing the resulting contours. Image obtained from photographs of salmon purchased from local fishery.

Figure 4 shows how the proposed method of image segmentation can successfully resolve the complex fin features of a fish, which is crucial when determining the surface area contribution of the fins. The outlined method above was used to normalize data acquisition for specimens with widely varied geometry. In all, 50 different specimens were used to assess the robustness of the image segmentation method.

### 3.1.2. NACA Airfoil Fit

This image segmentation process described in Section 3.1.1 was repeated for the top view to get the complete shape of the specimen. The top view of the fish and bio-robotic platforms resemble a symmetric NACA airfoil about the center line, as shown in Figure 6b. The approach of fitting the top to a NACA airfoil provides three useful benefits over fitting the top view to a polynomial. The first is that any asymmetry from the image is converted into a symmetrical shape about the center line of the animal. The second benefit is that the computational requirements for these calculations are less because calculations are only required for one half of the animal. Finally, if a certain animal or artificial swimmer is desired for computational fluid dynamic simulations, a 2D symmetric profile is readily available.

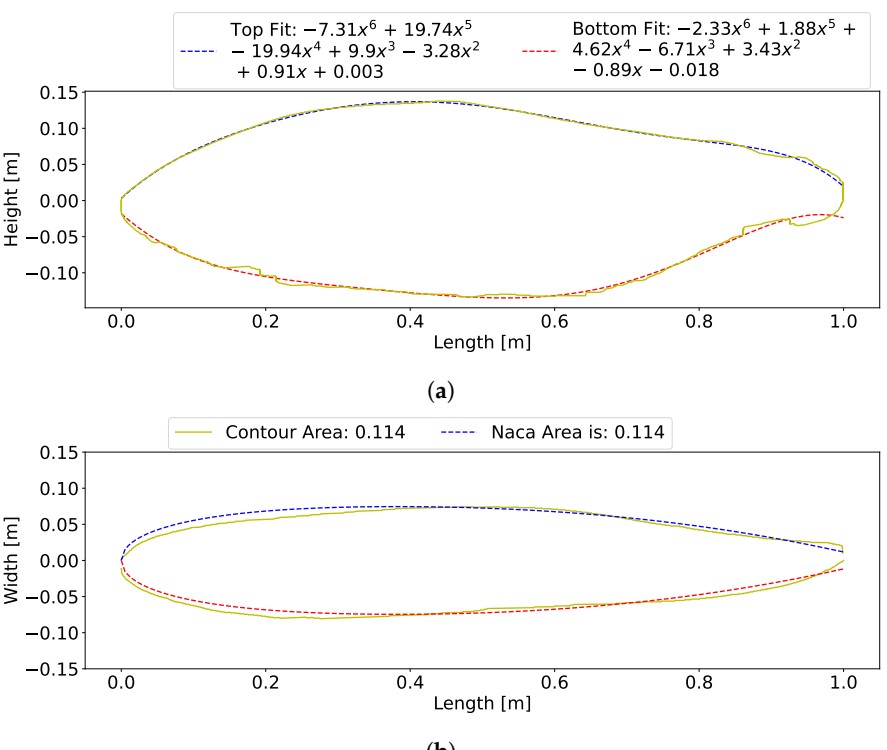

**Figure 6.** Contour lines (side and top views) of scanned salmon; the yellow line is the real contour line, solid (blue and red) lines are computed lines for side contour, and the blue dashed line is the computed line for top views. (**a**) King salmon side contour with fit equations shown in the legend. (**b**) King salmon top view contour with NACA fit overlayed. Area output from the area matching algorithm is shown.

NACA airfoils are given by an equation that is a function of chord length (the length of the foil from leading edge to trailing edge) and a percentage of the thickness of the chord. The default position of maximum thickness for a symmetric airfoil is at 30% of the chord length, but fish have varying locations of maximum thicknesses; therefore, a modified symmetrical NACA foil was used where the location of maximum thickness could be varied.

Normally, a symmetric NACA foil is represented by one equation, but the modified version is broken into a leading edge and trailing edge equation. For reference, a schematic is provided in Figure 7 and the equations used for the modified NACA airfoil is given by Charles et al. and Abbot in [47,48]:

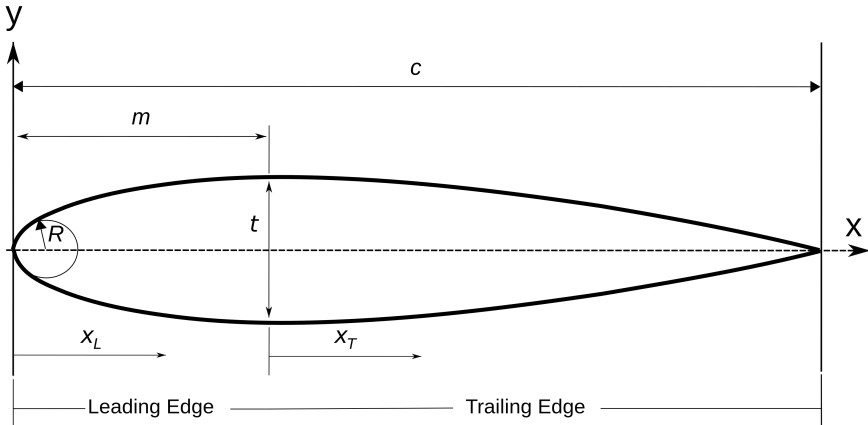

**Figure 7.** Schematic view of a symmetric airfoil. $c$ is the chord length, $t$ is the maximum thickness as a percentage of chord length, and $m$ is the x-position where maximum thickness occurs. $x_L$ represents the x-coordinate of the Leading Edge and $x_T$ is the coordinates of the Trailing Edge.

$$\bar{y}_{leading} = a_0\sqrt{\bar{x}_L} + a_1\bar{x}_L + a_2\bar{x}_L^2$$
$$+a_3\bar{x}_L^3, \tag{9}$$

$$\bar{y}_{trailing} = d_0 + d_1(1 - \bar{x}_T) + d_2(1 - \bar{x}_T)^2$$
$$+d_3(1 - \bar{x}_T)^3, \tag{10}$$

where $\bar{y}$ is the thickness over chord length $\left(\frac{y}{c}\right)$ and $\bar{x}_L$ and $\bar{x}_T$ is the $x$-position over chord length $\left(\frac{x}{c}\right)$ for the leading and trailing edge, respectively. The foil is a piece-wise combination of these two equations with the break point being the x-position of maximum thickness, $m$. In practice, the position is kept in tenths of chord length between 0.2 and 0.6. A survey of 33 fish contours was performed, and all fit within this interval, with the minimum value of 0.21 belonging to the ocean sunfish (Mola mola) and the maximum value of 0.48 belonging to the bowfin (Amia calva). In order to solve for the coefficients, the equations are subject to the following boundary conditions provided by Charles et al. and Abbot [47,48]:

For a maximum ordinate at $x = m$:

$$\bar{y} = \frac{t}{2c}, \qquad \left.\frac{d\bar{y}}{dx}\right|_{\bar{x}_L} = \left.\frac{d\bar{y}}{dx}\right|_{\bar{x}_T} = 0,$$
$$\left.\frac{d^2\bar{y}}{dx^2}\right|_{\bar{x}_L} = \left.\frac{d^2\bar{y}}{dx^2}\right|_{\bar{x}_T} \tag{11}$$

For a trailing edge ordinate at $(\bar{x}_T = 1.0)$:

$$\bar{y} = k, \tag{12}$$

where $k$ is half the size of the tail width given by the top view contour of the fish. This value defines the coefficient $d_0$.

The leading edge radius at $\bar{x}_L = 0$ is provided by Abbot [47]:

$$R = \frac{a_0^2}{2} = 1.1019\left(\frac{tI}{6}\right)^2, \tag{13}$$

where $I$ is an index number that determines the sharpness of the airfoil nose, 0 being sharp and 6 being a default value where the leading edge is round. This was kept to 6 for the algorithm in this research. The values of $d_1$ are chosen such that reversals in curvature are avoided using Table 2 [47,48]:

**Table 2.** Experimental values to define the coefficient $d_1$ as a function of m from Abbot [47].

| $m$ | $d_1$ |
|:---:|:---:|
| 0.2 | $1.000t$ |
| 0.3 | $1.170t$ |
| 0.4 | $1.575t$ |
| 0.5 | $3.325t$ |
| 0.6 | $3.500t$ |

Table 2 values are fit with a 3rd order polynomial so that a smooth interpolated value can be obtained for any max thickness, $m$, position. Coefficient $a_3$ is dependent on $d_2$ and $d_3$; therefore, $\bar{y}_{trailing}$ is solved before $\bar{y}_{leading}$. The coefficients are formed as a system of equations. $d_0$ and $d_1$ are given by the boundary conditions above, and $d_2$, $d_3$ are solved by:

$$\begin{bmatrix} (1-m)^2 & (1-m)^3 \\ -2(1-m) & -3(1-m)^2 \end{bmatrix} \begin{bmatrix} d_2 \\ d_3 \end{bmatrix} = \\ \begin{bmatrix} \dfrac{t}{2} - d_1(1-m) - d_0 \\ d_1 \end{bmatrix} \tag{14}$$

Coefficients $a_0$ are given by the boundary condition, and $a_1$, $a_2$, $a_3$ are solved by:

$$\begin{bmatrix} m & m^2 & m^3 \\ 1 & 2m & 3m^2 \\ 0 & 2 & 6m \end{bmatrix} \begin{bmatrix} a_1 \\ a_2 \\ a_3 \end{bmatrix} = \\ \begin{bmatrix} \dfrac{t}{2} - a_0 m^{\frac{1}{2}} \\ -\dfrac{a_0 m^{-\frac{1}{2}}}{2} \\ 2d_2 + 6d_3(1-m) + \frac{1}{4}a_0 m^{-\frac{3}{2}} \end{bmatrix} \tag{15}$$

Due to the irregular shape of the contour, a recursive function that minimizes the difference in area between the NACA foil and the contour is used to fit the thickness parameter. The function changes the airfoil thickness until a difference between the areas are within 0.001 of each other. Area is calculated using the Simpson integration procedure available through the SciPy package [49]. Figure 6 shows both the side and top contour and associated curve fits.

*3.2. Ellipsoid Approximation*

Three methods from the literature and the Ika-Fit method use different ellipsoid approximations to determine the surface area and volume of a specimen. The key difference between all these methods is in how the ellipse parameters are determined. Table 3 lists and briefly describes the methods and their respective references. Murphy and Haroutunian and Phillips et al. use the same formulation of an equivalent diameter for a prolate ellipsoid as a function of length and mass [3,12,13]:

**Table 3.** Abbreviation of methods, description of how the methods are implemented, and references.

| Method | Description | Reference |
|:---|:---:|:---:|
| Ika-Fit (IF) | Developed method using computer vision. | This paper |
| PDR | Ellipsoid partition disc method using computer vision | [14] |
| PSR | Prolate spheroid method method using computer vision | [14] |
| PSM | Prolate spheroid method using equivalent diameter | [3,12,13] |

Note: Blue background colors highlight the developed method's results.

$$D_e = \sqrt{\frac{6M}{\rho \pi L}}, \tag{16}$$

where $M$ is the mass of the fish in kilograms and $L$ is the total length of the subject in meters. This method is referred to as the prolate spheroid Murphy (PSM) method in this text. In contrast to the PSM method, Rantung et al. uses the measured width of the fish from their computer vision operations as the equivalent diameter, referred to as the prolate spheroid Rantung (PSR) method. The surface area can then be directly calculated using the equation for a prolate spheroid [3,12–14]:

$$SA = 2\pi \left(\frac{D_e}{2}\right)^2 + 2\pi \left(\frac{D_e L}{4e}\right) \sin e_p \tag{17}$$

$e_p$ is the prolate spheroid eccentricity defined as:

$$e_p = \sqrt{1 - \frac{D_e^2}{L^2}}. \tag{18}$$

Rantung et al. proposed a second method using an ellipse partition disc, which this text refers to as the partition disc Rantung (PDR) method. The fish geometry is composed as a series of partition discs as shown in Figure 8. The dimensions of two adjacent discs are defined as [14]:

$$
\begin{aligned}
a_{1i} &= \frac{W_{1i}}{2}, \quad a_{2i} = \frac{W_{2i}}{2}, \\
b_{1i} &= \frac{T_{1i}}{2}, \quad b_{2i} = \frac{T_{2i}}{2},
\end{aligned} \tag{19}
$$

where $W_{1i}$ and $T_{1i}$ are the body width and height at point $i$, and $dx_i$ is the thickness between discs 1 and 2, as shown in Figure 8. This is referred to as the partition disc Rantung method in this text. The surface area is calculated as [14]:

$$
\begin{aligned}
S_i &= \pi dx_i \left[\frac{(a_{1i} + b_{1i}) + (a_{2i} + b_{2i})}{2}\right] \; and \\
S &= \sum_i^{n/2} S_i \; for \; i = 2, 3, \ldots, n/2.
\end{aligned} \tag{20}
$$

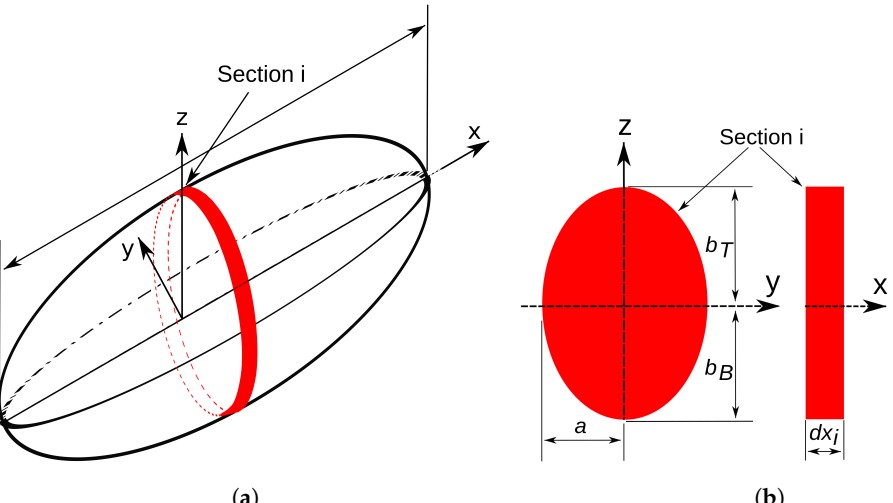

(a) (b)

**Figure 8.** (**a**) Prolate spheroid with elliptical section shown. Spheroid is partitioned into discs shown in red. (**b**) Schematic of elliptical section showing parameters used in surface area approximation: $b_T$ is the top contour, $b_B$ is the bottom contour, $a$ is the thickness of the NACA airfoil fit, and $dx$ is the thickness of the elliptical section.

In contrast with the existing methods discussed above, the Ika-Fit method uses the partition disc concept, but treats the top and bottom of the fish separately. The Ika-Fit method also uses the standard length of the subject versus the total length. The cross section of the specimen is assumed to be an ellipse. This is true for fusiform fish, but not the case for some other species. For elliptical cross-sections, an ellipse is fit as shown in Figure 8, where the top and bottom axis values at a certain z-axis point is given by the 6th degree poly fit of the side contour, $b_T$ and $b_B$. The top fit given by the NACA airfoil gives the semi-minor axis $a$ and the thickness of each disc is $dt$, as shown in Figure 8.

Consider a length along the perimeter of an ellipse:

$$ds = \sqrt{dx^2 + dy^2} = \sqrt{a^2 \cos^2 \phi + b^2 \sin^2 \phi}\, d\phi; \tag{21}$$

using the polar coordinate transform:

$$\begin{aligned} x &= a \sin \phi \\ y_T &= b_T \cos \phi \\ y_B &= b_B \cos \phi \end{aligned} \tag{22}$$

and defining eccentricity as:

$$e^2 = 1 - \frac{b_i^2}{a^2}, \tag{23}$$

where $b_i$ is $b_T$ or $b_B$ as described in Figure 8. Note that $b_i$ must be less than $a$ or $e^2$ is negative. Thus, the ellipse is oriented such that $e^2$ is a positive value. Substituting this into $ds$ gives:

$$ds = a\sqrt{1 - e^2 \sin^2 \phi}\, d\phi \tag{24}$$

Natural and artificial fish subjects are assumed to be planarly symmetric, and so only 1/4 of the ellipse is needed for the top section and bottom section. Integrating $ds$ from 0 to $\pi/2$ gives:

$$\begin{aligned} \int ds = a \int_0^{\frac{\pi}{2}} \sqrt{1 - e^2 \sin^2 \phi}\, d\phi = \\ a\left[ E\left(e, \frac{\pi}{2}\right) - E(e, 0) \right] \end{aligned} \tag{25}$$

The SciPy package is used to solve this integral and it approximates the solution as follows [49]:

$$E(e) \approx P(1 - e) - (1 - e)\log(1 - e)Q(1 - e), \tag{26}$$

where $P$ and $Q$ are 10th degree polynomials. To convert the perimeter to a surface area, the $x$-components are evenly spaced and the difference in each $x$ value is taken as the thickness. For (N-1) points, the perimeters of the ellipse are multiplied by the thickness and the sum is taken. This differs from the PDR approach in that the solution they use for the perimeter is an infinite binomial series with higher order terms vanishing to zero and $a_i$ and $b_i$ defined in (19) [14]:

$$p = \pi(a_i + b_i) \tag{27}$$

A further difference is that the PDR method is for half the subject. The Ika-Fit method uses the NACA airfoil as the top view contour and divides the ellipse into a sum of two different ellipses, with the top view contour determining the $a$, and $b_T$ and $b_B$ being the difference in height from the $x$-axis to the side view contour.

To include the fins in the surface area, the contour area is calculated on images with and without fins intact using the OpenCV package and Green's functions [45]. The two areas are divided to get an area ratio of fins to no fins. The area ratio is added to the calculated surface area as a percentage of the total surface area. Output from the algorithm is provided for biological animals in Table S1: Biological Fit Data and for robotic platforms in Table S2: Robot Fit and CoT Data.

## 4. Results

### 4.1. Validation with 3D Scanned Model

The 3D mesh of the scanned salmon shown in Figure 2 was imported into the open-source software Blender and dimensions were consistently scaled from lengths of 0.1 to 100 m. Surface area and volume were evaluated within Blender software using the 3D print add-on. Validation of the surface area and volume are given in Figure 9.

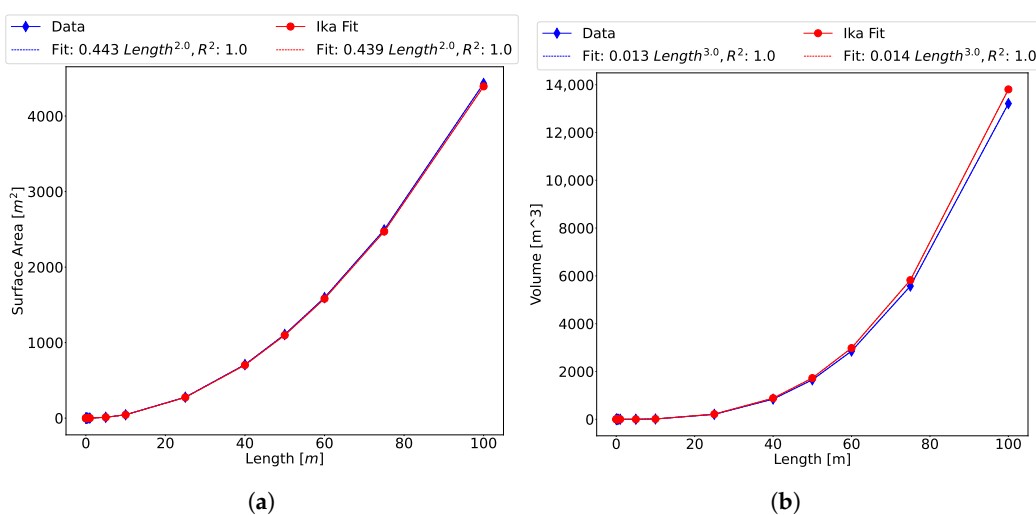

**Figure 9.** (**a**,**b**) Validation of Ika-Fit (**a**) surface area and (**b**) volume algorithm with scanned salmon data scaled using Blender software.

The figures show a good agreement with the measured data through a large interval of length scales. This data further validates the method's ability to scale surface area and volume with the model's length. An important finding is that the volume also scales within good agreement, which allows for the calculation of mass by assuming that the animal or robot is neutrally buoyant in water. Then, the mass property is deduced:

$$M_{platform} = \rho_{water} \cdot V_{platform},\tag{28}$$

where $\rho_{water}$ is taken to be the average density of water, $1025\frac{\text{kg}}{\text{m}^3}$. A dissection of fish by Haroutunian [12] shows that the fins account for approximately 1% of the total mass of the fish, so they are excluded from the volume calculation.

The error between different datasets is reported as root mean square difference (RMSD) and mean absolute error (MAE) using the following equations:

$$RMSD = \sqrt{\frac{\sum_{n=1}^{N}(\hat{y}_t - y_t)^2}{N}},\tag{29}$$

$$MAE = \frac{\sum_{n=1}^{N}|\hat{y}_t - y_t|}{N},\tag{30}$$

where $\hat{y}_t$ is the estimated value, $y_t$ is the published data, and $N$ is the number of data points. This formulation is used for the following analysis and in the rest of this paper.

### 4.2. Validation with Biological Animals

Four species were used in the validation of the mass: Atlantic salmon (Salmo Salar), Atlantic cod (Gadus Morhua), killer whale (Orcinus orca), and European silver eel (Anguilla Anguilla). Data from O'Shea et al. [22] were used for the Atlantic salmon and cod. For the European silver eel, the length-weight relationship from Froese and Pauly [50] was used. The length-weight relationship used for the killer whale is given in Bigg and Wolman [51]. No images were available for the killer whale or silver eel, so a 3D model was constructed in the open source software Blender based on reference pictures. Figure 10 shows the mass calculated using the Ika-Fit method compared to published data.

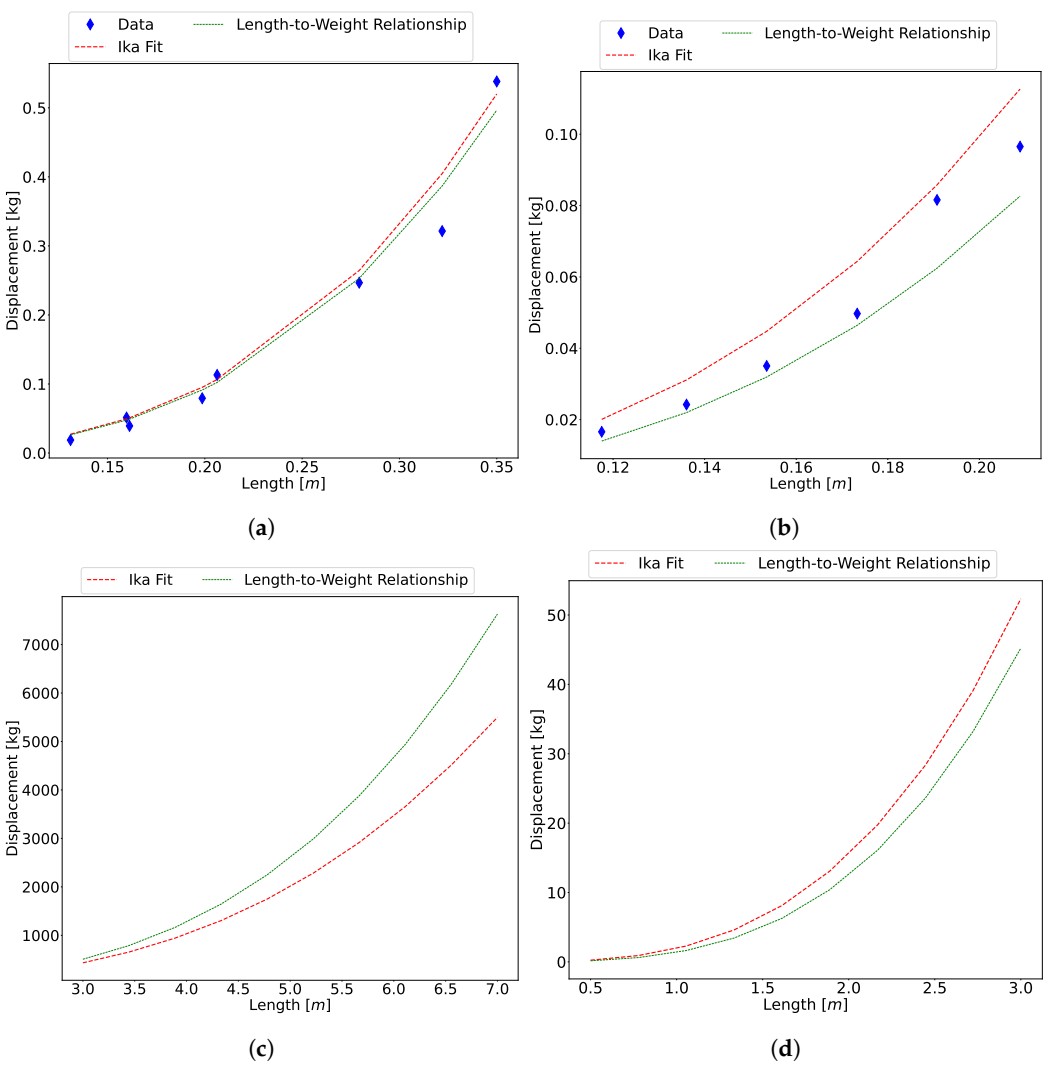

**Figure 10.** (**a**) The calculated mass versus published data for (**a**) Atlantic salmon, (**b**) Atlantic cod, (**c**) killer whale, and (**d**) European silver eel.

The figure shows that there is good agreement between the data and the estimation from the algorithm. Table 4 shows the error of the Ika-Fit method with the length-weight relationships of each species.

**Table 4.** RMSD and MAE of the Ika-Fit method of four animal species.

| Error | Salmon | Cod | Killer Whale | European Silver Eel |
|-------|--------|-----|--------------|---------------------|
| RMSD | 0.032 | 0.010 | 172.328 | 3.655 |
| MAE | 0.020 | 0.009 | 161.478 | 2.810 |

These species were chosen according to the available data of both physical morphology and CoT. These results further show the flexibility of the proposed method. A 3D model can be constructed from reference images, and the results are similar to published data of multiple different samples of the same species.

### 4.2.1. Comparison with Other Algorithms

As discussed in Section 3.2 and shown in Table 3, there are three other methods that our results are compared against. Figure 11 shows the comparison between the methods on the 3D scanned model. The Ika-Fit method shows better agreement with the scaled data than comparison methods. For reference, a power-law fit for each method is shown, as is convention with allometric data, and the error for each method is shown in Table 5.

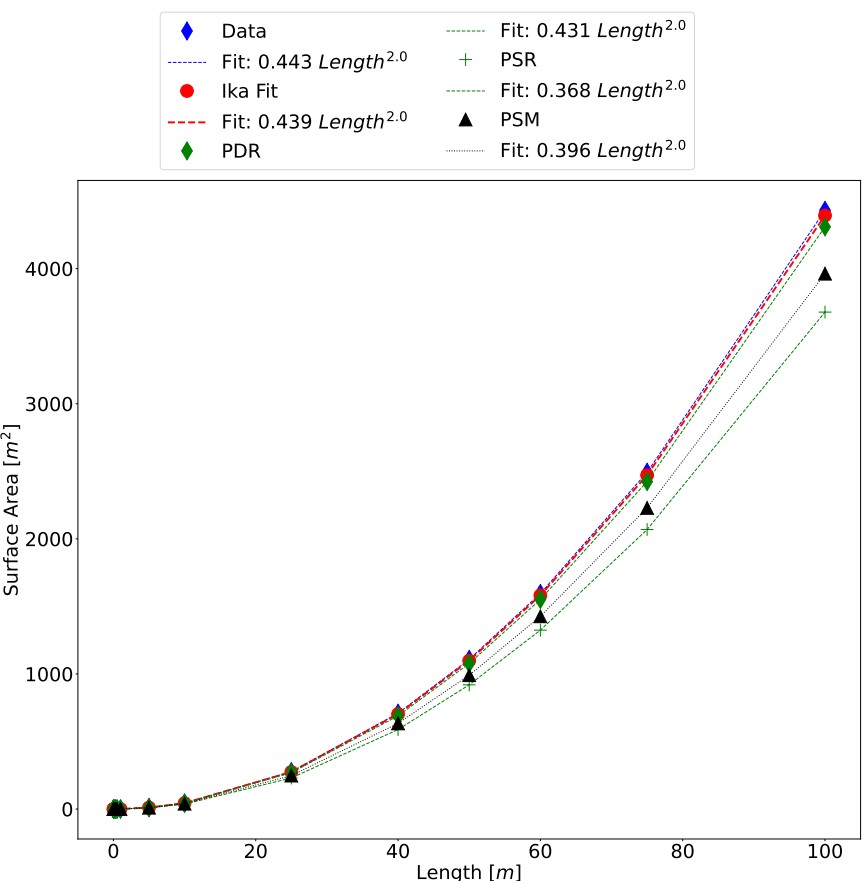

**Figure 11.** Comparison of surface area of all algorithms with the scanned salmon model that has been scaled inside Blender.

**Table 5.** RMSD and MAE of compared methods with laser-scanned salmon.

| Error | IF | PDR | PSR | PSM |
|:---:|:---:|:---:|:---:|:---:|
| RMSD | 12.25 | 38.98 | 241.13 | 150.10 |
| MAE | 6.14 | 19.54 | 120.87 | 75.23 |

Note: Blue background colors highlight the developed method's results.

The comparison shows that the developed Ika-Fit method outperforms the other methods in terms of accuracy. The PDR method gives the next best estimation of surface area. These two methods are similar in that they both partition the specimen into ellipse discs, with the difference being the NACA fit of the top of the specimen, as well as the inclusion of fin surface area.

To compare all methods to biological data for surface area, data from O'Shea et al. were used with images of Atlantic cod (Gadus morhua) and Atlantic salmon (Salmo salar) [22].

A further validation is presented with a high precision 3D scan of a king salmon. Due to the king salmon and Atlantic salmon not being the same species but part of the same family, a scaling factor was formed to scale the contours to be closer to the Atlantic salmon. No top images of Atlantic salmon were available, nor was a real fish available for purchase and laser scanning for proper dimensions.

For comparison with published methods, the PDR, PSR, and PSM methods are included in the analysis. It should be noted that the PDR and PSR methods use the contour data from the computer vision algorithm, since height and width data was not reported in the published research. The mass needed for the PSM method was directly taken from the data given in [22], since those values were reported. Tables 6 and 7 give the relative error of all approximation methods with the data of O'Shea et al.

**Table 6.** RMSD and MAE of compared methods with Atlantic salmon data from O'Shea et al. [22].

| Error | IF | PDR | PSR | PSM |
|---|---|---|---|---|
| RMSD | 0.0025 | 0.0035 | 0.0066 | 0.0082 |
| MAE | 0.0016 | 0.0029 | 0.0057 | 0.0074 |

Note: Blue background colors highlight the developed method's results.

**Table 7.** RMSD and MAE of compared methods with Atlantic cod data from O'Shea et al. [22].

| Error | IF | PD | PSR | PSM |
|---|---|---|---|---|
| RMSD | 0.0013 | 0.0024 | 0.0010 | 0.0050 |
| MAE | 0.0011 | 0.0020 | 0.0009 | 0.0045 |

Note: Blue background colors highlight the developed method's results.

To determine the effect of the fin surface area, the same fin area ratio was added to the final surface area of the other methods. Figure 12 shows that the addition of the fin area improves the accuracy of all the other methods.

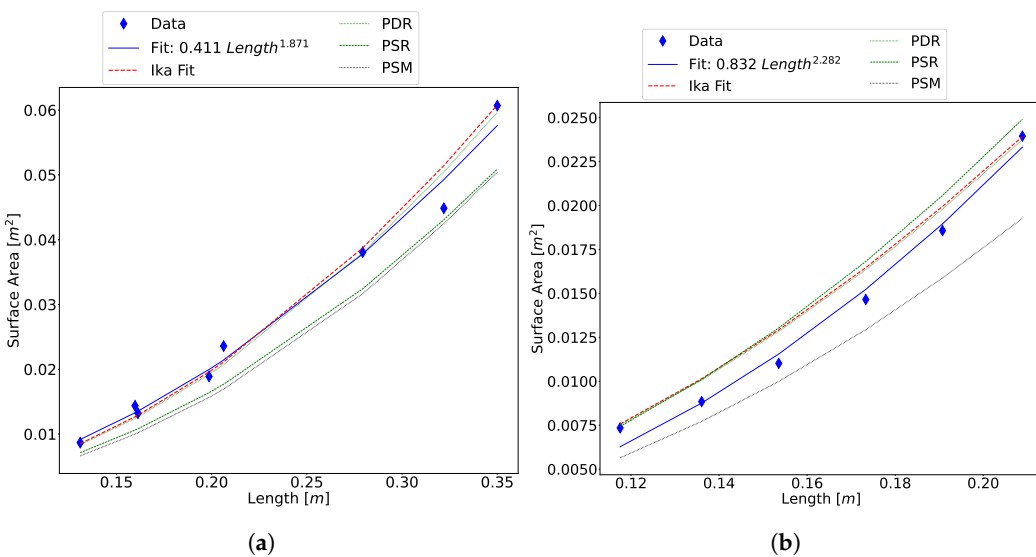

(**a**)               (**b**)

**Figure 12.** Comparison of algorithms with the fin area added to the final surface area. (**a**) Scaled Atlantic salmon data and (**b**) Atlantic cod data.

The data shows that the Ika-Fit method and the PDR perform similarly to each other when the fin area ratio is added. This is expected, as the difference between the two methods is treating the top and bottom of the animal as separate partitions, as well as the NACA airfoil fit and the inclusion of the fin area ratio. The PSR and PSM methods performs well in the case of the Atlantic cod, but is not consistent when comparing it to the Atlantic salmon data.

To determine the breakdown of the two prolate spheroid methods, the equivalent diameter was calculated and superimposed on the side and top view contours of the king salmon. As shown in Figure 13, the equivalent diameter underestimates the side and top contour. This leads to the prolate spheroid surface area being underestimated, which is clearly shown in the comparison data of Figures 12 and 14.

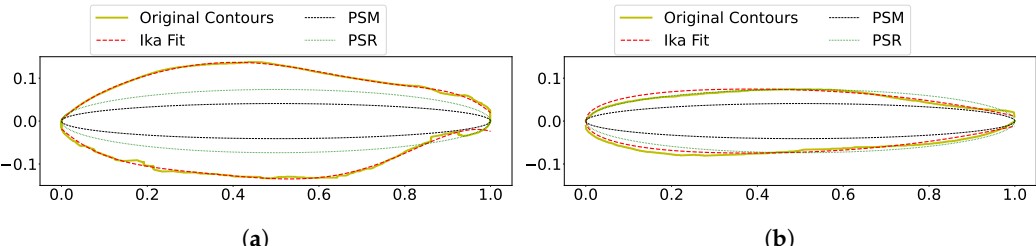

(**a**)           (**b**)

**Figure 13.** Equivalent diameter ellipse superimposed on fit contour. (**a**) Side and (**b**) top views of king salmon. Width used as input into the ellipse of Rantung et al. was taken from the top contour output from the computer vision operations and not from the NACA airfoil fit.

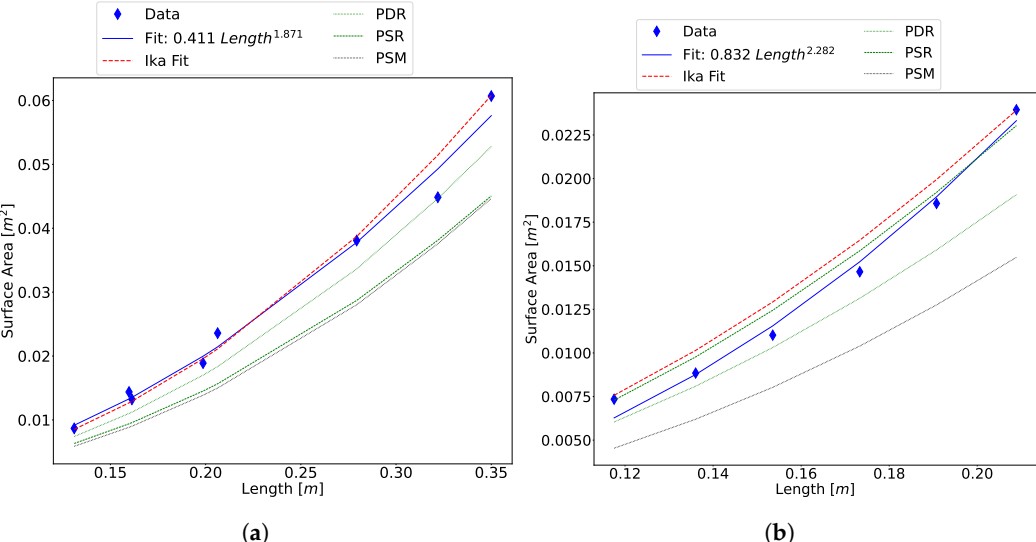

(**a**)           (**b**)

**Figure 14.** Comparison plot between the surface area estimation algorithm, prolate spheroid method [4,14], partition disc method [14], and O'Shea regression lines [22]. (**a**) Unscaled Atlantic salmon data, (**b**) scaled Atlantic salmon data.

A study was performed to determine the relative time difference between the wrap method presented by O'Shea et al. [22] and the comparison algorithms. Table 8 lists the time it takes for each algorithm to perform operations in the following order: importing images, performing computer vision operations, fitting all contours, finding minimum and maximum points, fitting the NACA airfoil, and calculating the surface area. The PSM method does not require computer vision operations; therefore, the time for this is defining variables and estimating the surface area.

**Table 8.** Relative time difference between methods. Time was measured from the image input to the value output.

| Method | Time Difference [s] |
| --- | --- |
| O'Shea et al. wrap method | laborious |
| Ika-Fit | 1.3925 |
| PD | 1.3540 |
| PSR | 1.3384 |
| PSM | 0.0004 |

Note: Blue background colors highlight the developed method's results.

4.2.2. Robotic Data

To extend the algorithm to bio-inspired robotic models, a database of 81 conventional AUVs and 139 bio-robots was compiled. Of the bio-inspired robots, 80 operate with body-caudal fin (BCF) propulsion, 35 use median-paired fin (MPF) swimming mode, 13 use lift-based propulsion, and the remaining robots are inspired by other propulsion modes. Out of the 139 bio-inspired robots, only 4 reported their surface area, and their relevant parameters are summarized in Table 9.

**Table 9.** Bio-inspired platforms used as validation in this research. Relevant parameters and citations are given.

| Platform Name | Total Length [m] | Standard Length [m] | Width [m] |
|---|---|---|---|
| Harvard Beihang Mackerel [52] | 0.588 | 0.548 | 0.080 |
| MIT Carangiform [53] | 0.148 | 0.127 | 0.025 |
| NYU iDevice [54] | 0.066 | 0.0452 | 0.019 |
| NRL 4-Fin [55] | 0.438 | 0.438 | 0.089 |
| | **Height [m]** | **Mass [kg]** | **Surface Area [m$^2$]** |
| Harvard Beihang Mackerel [52] | 0.95 | 2.79 | 0.137 |
| MIT Carangiform [53] | 0.043 | 0.068 | 0.013 |
| NYU iDevice [54] | 0.021 | 0.009 | 0.0032 |
| NRL 4-Fin [55] | 0.089 | 0.178 | 2.9 |

The results of the Ika-Fit method on these bio-robotic models is shown in Figure 15. The Figure shows that the Ika-Fit method performs the same as or better than other methods. Interestingly, the PSM method performs poorly for the last two bio-robotic models. An explanation of this is due to the formulation of the equivalent diameter, which is only a function of length and mass of the animal/robot. In the case of the Naval Research Laboratories four-fin platform, the mass is close to that of the Harvard Beihang Mackerel but the surface area is nearly double.

The mass of the robots were calculated by multiplying the volume by water density. The PSM method performed the best due to the method formulation that guarantees the same mass is output when multiplied by the density of water. This test highlights a limitation of the Ika-Fit method when applied to engineered systems like bio-mimetic robots. In cases where the designers stayed as close to the biological analogue as possible, as in [52,53], the Ika-Fit method performs the best. In cases where the biological analogue is not followed, as in [54,55], the Ika-Fit method doesn't perform well in estimating mass. This can be resolved by calculating an equivalent mass factor based on the published and calculated mass that can account for the different design parameters, such as:

$$M_{eq} = \frac{M_{published}}{M_{calculated}} \tag{31}$$

*4.3. Application to Cost of Transport*

The primary purpose of an accurate parameter measurement is the application to CoT in (5) to obtain an estimation of the propulsion power. This is only half of the CoT equation, and the hotel power must also be estimated. For the purpose of the following analysis, the hotel power of biological animals is estimated using data collected by [3], and a power-law curve fit is applied to obtain an allometric relationship, as shown in Figure 16.

The relationships derived from this data are used in conjunction with the Ika-Fit method and the PSM method outlined in [3] to compare the derived CoT model against aquatic animals. The PSM method is included to give a comparison as to the relative importance of the estimation method for physical parameters of the animal within the complete CoT formulation. Figure 17 shows that the two methods converge in region 1

and slowly diverge as velocities increase. This is to be expected as hotel power dominates the model in this region, and so the estimation of physical parameters is less important. As the propulsive power dominates in region 2, the data shows that the two estimations diverge and do not follow the same trend lines. This is particularly apparent in the data for the European silver eel, where the Ika-Fit method seems to follow the data more closely than the PSM method.

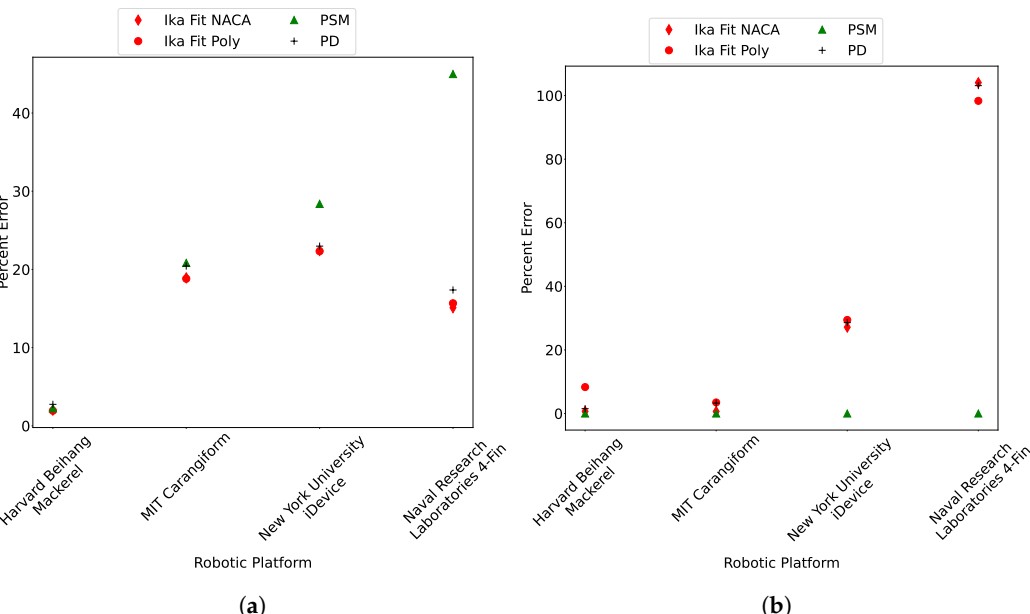

**Figure 15.** (**a**) Comparison of surface area and (**b**) mass for bio-robotic platforms. Ika-Fit NACA and Ika-Fit Poly are this study's method using the NACA fit described in Section 3.1.2, and Ika-Fit Poly is the top contour's fit with a fourth degree polynomial in a similar way to the side contours.

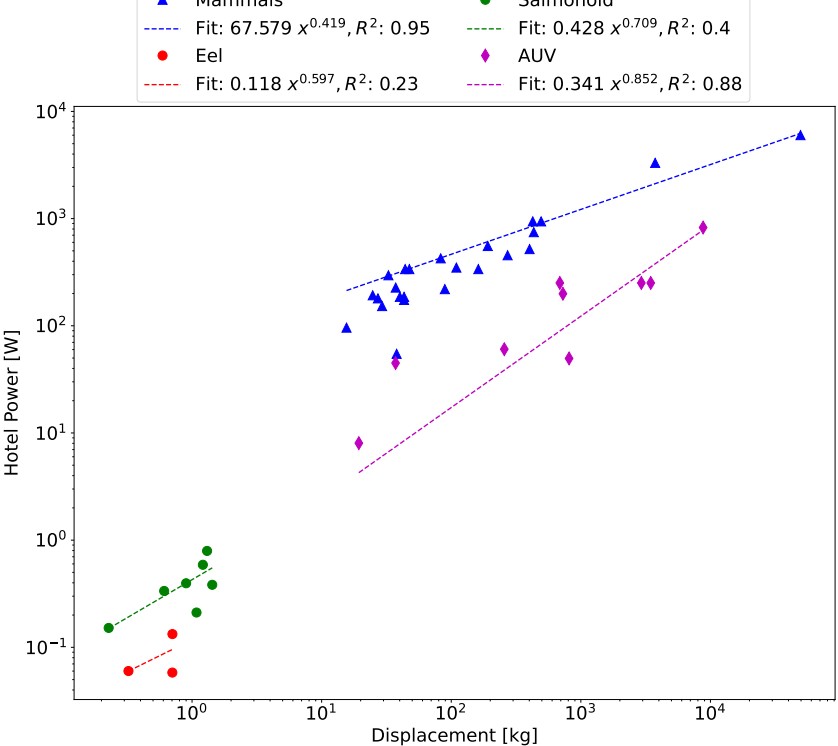

**Figure 16.** Data collected in [3] fit with a power-law curve to obtain the hotel power for biological animals.

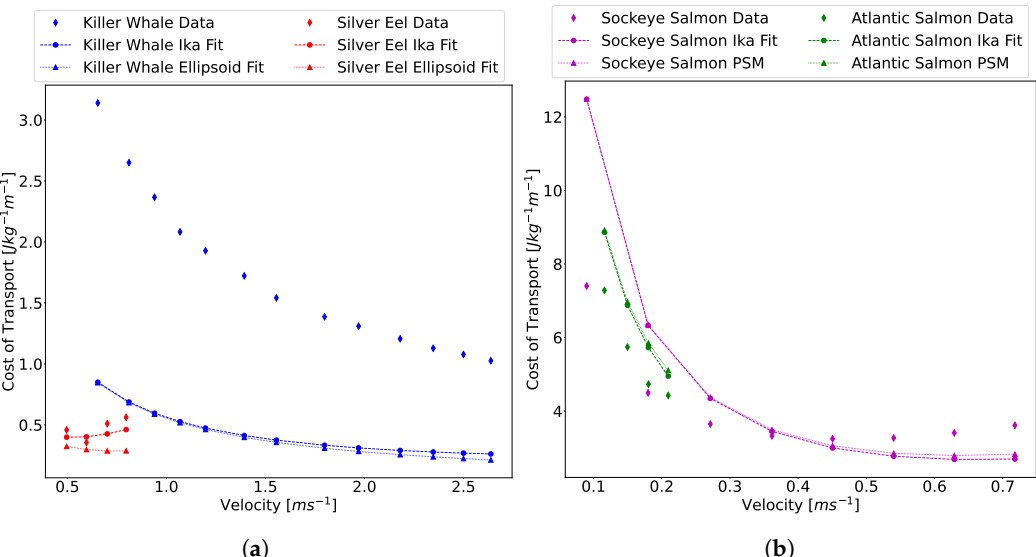

**Figure 17.** Comparison of the Ika-Fit and PSM method in calculating CoT with published data. (**a**) Killer whale data from [56] and European silver eel data from [57]. (**b**) Sockeye salmon data from [16] and Atlantic salmon data from [8].

There is a significant difference in the hotel power dominant region for the two salmonoid specimens. The method used to calculate hotel power was the power-law fit for salmonoid data from Figure 16 and the results suggest that a hotel power for each specific species should be derived separately from each other. This also shows the importance of an accurate hotel power estimate for animals and vehicles operating in region 1, as there is a significant gap between the data and calculated data as hotel power becomes dominant.

There is a significant difference between the data and the model of (5) at higher velocities, as is most clearly seen in the data for the killer whale specimen. As an animal or robot swims, there are three sources of drag, as explained by Magnuson [18]: friction and form drag, gill resistance, and induced drag. The ITTC method, explained in Section 1 and used to determine $C_D$ in (5), only gives the friction and form drag. This is synonymous with the towed resistance of the animal/vehicle. Data estimated on skipjack tuna swimming at sustained speeds of 66 cms$^{-1}$ showed that induced drag accounts for 30% and gill resistance accounts for 17% of total drag [18]. The induced drag arises from wake effects and shed vortexes from the propulsor interacting with the water.

The biological data itself has some assumptions built into it that may contribute to the disparity in data. Specifically, the killer whale data is of wild animals swimming in open water. As noted by the author, the calculations assumed straight line swimming, and swim speeds were not corrected for the effect of tidal currents. These could not be made, since the orientation of the whale with respect to the tidal currents was hard to determine [56].

In an effort to limit uncertainty from the analysis and to test the efficacy of the CoT model, the Ika-Fit method was compared to bio-robotic data. This approach was taken because it removes most uncertainties due to electrical power output, or the combination of hotel power and propulsion power, being measured exactly, as well as velocity, mass, and physical dimensions. To determine hotel power for these robots, total power is plotted at each velocity. The data is then extrapolated with a curve fit to velocity $U = 0$. The total power at velocity $U = 0$ is determined to be the hotel power of the artificial swimmer. This is analogous to what is done for biological swimmers [16]. Out of the 139 bio-mimetic robots surveyed, 3 groups reported CoT results and the comparison is shown in Figure 18. The RoboSalmon and Knifebot both reported total power for a range of velocities. The UVTunabot only reported CoT, and total power was calculated by multiplying CoT by mass and velocity at that point. The extrapolation method described above was used in all cases.

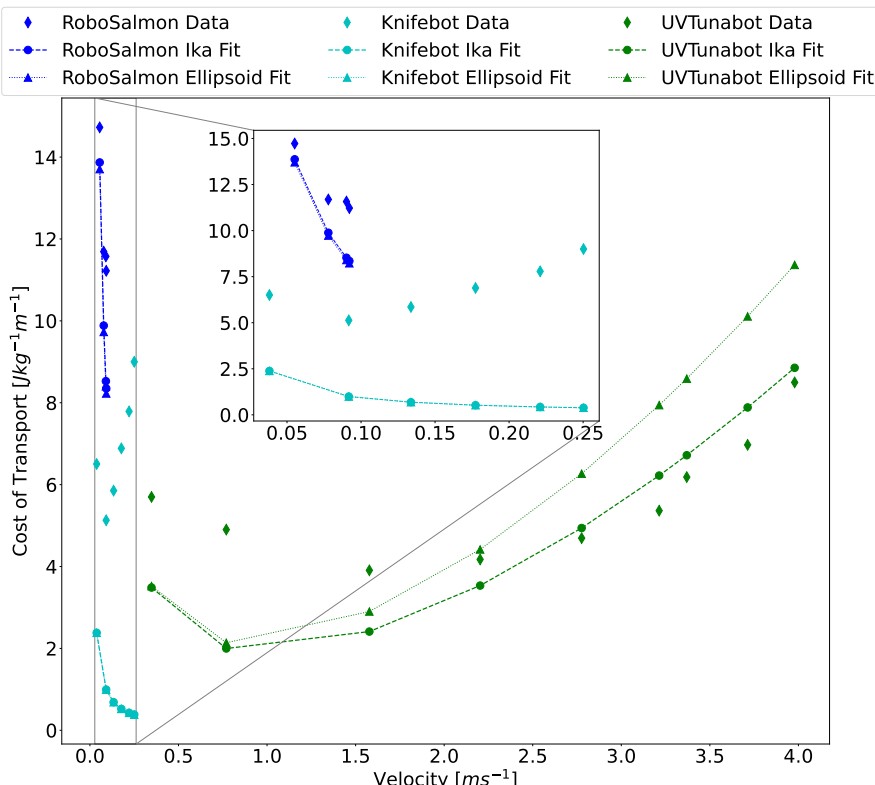

**Figure 18.** Comparison of the Ika-Fit and PSM method in calculating CoT with published bio-mimetic robot data. RoboSalmon data from [58], Knifebot data from [59], and UVTunabot data from [60]. The dashed black line marks the demarcation of regions 1 and 2 for the UVTunabot.

It is important to note the different modes of locomotion encompassed by the data and how it relates to the CoT model. The RoboSalmon operates in sub-carangiform mode, which means that the rear 50% of the entire body undergoes undulation. The UVTunabot operates in thunniform locomotion, in which only the tail or aft third of the fish undergoes oscillation. The Knifebot uses gymnotiform locomotion similar to that of a black ghost knifefish. This robot has a fin that runs the length of the body and oscillates in a sinusoidal pattern to create locomotion.

Data in Figure 18 shows that the developed method and CoT model have the same drawbacks, as discussed in Figure 17. The published CoT is significantly greater than the modeled CoT for the non-thunniform swimmers. The UVTunabot tracks closely with the model, because the body of the robot is mostly rigid while swimming. Interestingly, there is a cross-over point in the UVTunabot data in region 2 at around 2.25 ms$^{-1}$ that is not present in the other bio-robotic platforms or the biological data comparison of Figure 17. It is expected that the model of (5) would underestimate values in region 2 as explained above, but that is not the case here. Comparing the velocities of the robot data and biological data of Figure 17 shows that the highest biological speed reported is around 2.75 ms$^{-1}$.

Another consideration is the efficiency of the actuator and propulsor, which varies with velocity and was reported only for [58,59]. For the UVTunabot data shown in Figure 18, the actuation and propulsion efficiency for the UVTunabot was estimated to be 0.64, with the DC motor having an efficiency of 0.8 and the propulsor having an efficiency of 0.8. Propulsor efficiency was chosen because the reported Strouhal number is between 0.2–0.4, which is shown to reach propulsion efficiencies of up to 0.9 [43]. The actuation mechanism was a DC motor that has efficiencies in the range of 0.6–0.9, as shown in Table 1. As seen in Figure 1d, efficiencies have the marked effect of either flattening or rounding the CoT curve in region 2.

Comparing the Ika-Fit algorithm to the PSM algorithm for the bio-robotic data, there is a significant divergence for the UVTunabot data; this divergence is less so for the other

two platforms. Out of the three robots shown, the UVTunabot is the least like a prolate spheroid, with the max height being more than 50% the length. This implies that the PSM method is overestimating the surface area and shows how the approach taken by the Ika-Fit method is more accurate.

The results outlined above show how sensitive the CoT model is to input parameters, as well as the fact that the CoT model lacks the formulation to include complex induced drag and wake effects. The model works well when less of the body is undergoing undulation for propulsion, which suggests that it is a good model to use when evaluating ships, AUVs, and submarines, as was its original purpose. To apply this model to biological animals or bio-mimetic robots, there needs to be further research into propulsion power at different swimming speeds, and therefore different Reynolds numbers, for the various modes of locomotion.

For input parameters to (5), the Ika-Fit method performs more accurately than comparison methods at estimating physical parameters, including the surface area, volume, mass, and slenderness ratio. This method has the added benefit of not requiring a physical specimen and performs the estimation with only the length of the specimen as an input parameter. This is beneficial because it allows for the scaling of a platform in order to give an estimate of the CoT at different length scales or Reynolds numbers. This is further useful if the only information provided about a platform is the length.

A limitation of the Ika-Fit method comes from the estimation of mass from the volume and density of a specimen. When estimating the mass of engineered systems such as bio-robotics vehicles, as in Figure 15b, the estimated mass doesn't perform well in specific use cases when there is a large discrepancy between the lumped mass and center of gravity versus a more natural mass distribution. This implies that the platform is larger than it needs to be, but such a platform may be needed based on payload requirements. Referring to Table 9, it can be seen that the Harvard Beihang Mackerel and the NRL four-fin have similar lengths and mass, but the surface area, and volume by analogy, of the NRL four-fin is almost twice as much as the Harvard Beihang Mackerel. This would cause the mass estimation to increase, since it is based on volume.

Another limitation to the proposed algorithm is the need for two views of the platform for an accurate measurement of the platform's parameters. In many publications, there is a lack of orthogonal views of the platform. The Ika-Fit method is limited by assuming the "unknown" view by either estimating the view with a NACA profile or constructing a 3D model based on other reference pictures, as done with the European silver eel in Section 4. A summary of pros and cons of the Ika-Fit method is given in Table 10.

**Table 10.** Pros and Cons of the using the Ika-Fit method.

| | |
|---|---|
| Pros | Accurate estimation of physical parameters |
| | Good estimation of mass based on volume and density |
| | Ability to get an estimate at any length scale |
| | Only length needed with no other knowledge or physical specimen needed |
| Cons | Needs side and top views of platform |
| | Not accurate for a large mismatch between size and mass |
| | Cross section of platform needs to be circular or elliptical-like |

When the Ika-Fit method is applied to (5), there is a gap between published and calculated data. This is caused by (5) not including other forms of drag present in swimming animals. The cross-over point at higher velocities is an area of further research in order to formulate a more accurate model for propulsion power. Another finding is that the CoT model is very sensitive to hotel power, and so a better estimation for unknown quantities is needed. Shortcomings aside, (5) gives a consistent and easy to calculate method to compare with published CoT data for natural and artificial swimmers. Further research needs to be done to develop a more accurate CoT model that addresses the discrepancies found in this research.

## 5. Conclusions

In this work, a new methodology for determining physical parameters and CoT of artificial and natural swimmers when no CoT data exists is developed by extending the core CoT model of Phillips et al. [3,4]. The developed Ika-Fit method uses image segmentation techniques and ellipsoid estimations to estimate the surface area and volume of engineered systems. These parameters are used to estimate the CoT using a simplified model.

The Ika-Fit method shows good accuracy when measuring the surface area, volume, and mass of fish species and fish-shaped robotic platforms. An accuracy of within 20% for surface area for engineered systems that are not fish-shaped can also be obtained. The estimated CoT is shown to underestimate when compared to real-world data. This can be explained partly by the CoT model used not including induced drag from wake effects. More research is needed to improve the CoT model to reflect real-world data.

This method is useful to a designer of underwater vehicles when mission time and energy efficiency are important factors for specific applications. The CoT for many different platforms can be estimated to obtain a relative energy efficiency in the design phase. The comparison of CoT can elucidate whether it is beneficial to design a fish-like underwater vehicle or a propeller-driven AUV for long distance missions. This method can also be useful for biologists who want a way to perform a non-contact estimation of cost of transport for natural swimmers.

**Supplementary Materials:** The following are available at https://www.mdpi.com/article/10.3390/designs5040069/s1, Table S1: Biological Fit Data, Table S2: Robot Fit and CoT Data.

**Author Contributions:** Conceptualization, M.C. and S.G.; methodology, M.C.; software, M.C.; validation, M.C. and S.G.; formal analysis, M.C.; investigation, M.C.; resources, S.G.; data curation, M.C.; writing—original draft preparation, M.C.; writing—review and editing, S.G.; visualization, M.C.; supervision, S.G.; project administration, S.G.; funding acquisition, S.G. All authors have read and agreed to the published version of the manuscript.

**Funding:** This research received no external funding.

**Institutional Review Board Statement:** Not applicable.

**Informed Consent Statement:** Not applicable.

**Data Availability Statement:** All code scripts are available at https://github.com/michaelcoe/Ika-Fit (accessed on 2 September 2021).

**Acknowledgments:** The authors would like to acknowledge the support of the Mechanical Engineering Department at the University of Canterbury for funding this research.

**Conflicts of Interest:** The authors declare no conflict of interest.

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
