# Peer review of "Computer Vision Estimation of Physical Parameters and Its Application to Power Requirements of Natural and Artificial Swimmers"

_designs, 2020_

Round 1
Reviewer 1 Report
In this manuscript, the authors proposed a computer vision-based method, Ika-Fit to estimate physical characteristics of natural and artificial swimmers.
The introduction part is too long. Just reduce the length and focus on the key problems. Background information related to cost of transport can be shorted. If it is really necessary, put an additional section just after Introduction section.
The key steps are how to use image processing (image segmentation) to extract the contours and use a fitting method to fit the shape of the inspection subjects. For the segmentation part, the method is not new and more advanced segmentation algorithms using deep learning seems to be better. The NACA airfoil fit method is good but already exists. One more concern is that the authors didn’t give any example of the real application.
Reviewer 2 Report
The article may be of interest to a certain circle of readers, but it is necessary to eliminate a number of problematic aspects.
Intoduction section is completely changed. Show the readers new and old developments on the problem. Indicate who is involved in this. Where is the analysis and formulation of the research problem? No description of critical areas. There is no critical comparison of literary sources.
Mathematical formulas are poorly substantiated. The simulation environment needs to be justified. Figure 17. needs to be presented differently, more clearly and informatively. Why are the 2020 and 2021 publications from the Q / Q2 level journals not well represented?
Round 2
Reviewer 1 Report
The authors have addressed all my questions.
Reviewer 2 Report
The work is interesting. The authors worked through the comments well and the article became better and more interesting. The formulas are correct. The graphs are explained.